

**Triple collocation validates CONUS-wide evapotranspiration inferred from atmospheric**
**conditions**
Erica L. McCormick[1,*], Lillian E. Sanders[1], Kaighin A. McColl[2,3], Alexandra G. Konings[1]
[1] Department of Earth System Science, Stanford University
[2] Department of Earth and Planetary Sciences, Harvard University
[3] School of Engineering and Applied Sciences, Harvard University
*Corresponding author: ericamcc@stanford.edu
**Abstract**
Large-scale estimation of evapotranspiration (ET) remains challenging because no direct remote
sensing estimates of ET exist and because most data-driven estimation approaches require
assumptions about the impact of moisture conditions and biogeography on ET. The surface flux
equilibrium (SFE) approach offers an alternative, deriving ET directly from atmospheric
temperature and humidity under the assumption that conditions in the atmospheric boundary
layer reflect ET's land boundary condition. We present a 4 km resolution, continental United
States-wide, daily ET dataset spanning from 1979 to 2024 using the SFE method. The Bowen
ratio is first calculated using the SFE method solely based on temperature and specific humidity
estimates from gridMET and then converted to ET using net radiation and ground heat fluxes
from ERA5-Land. We evaluate its performance using extended triple collocation to estimate the
standard deviation of the random error and the correlation coefficient of SFE ET compared to
true ET, as well as those of three widely used alternative ET datasets: GLEAM, FluxCom, and
ERA5-Land. Despite its extreme simplicity, SFE ET achieves performance comparable to or
exceeding the other datasets across large portions of CONUS, particularly in the Western U.S.,
while requiring no information about land surface, vegetation, or soil properties and no
assumptions about ET's response to environmental and climate drivers. Our results support the
use of SFE as a scalable, observation-driven method for estimating ET.
**1. Introduction**
Evapotranspiration (ET) dominates the terrestrial water cycle (Friedlingstein et al., 2019; Good
et al., 2015), controls the partitioning of radiation into latent and sensible heat (McColl and
Rigden, 2020), and plays a key role in driving the hydrologic cycle by returning water to the
atmosphere (Oki and Kanae, 2006). ET therefore has downstream feedbacks on temperature



(Teuling et al., 2010), precipitation, and vegetation productivity (Green et al., 2017) in addition
to directly impacting the carbon cycles through the trade-off between photosynthesis and
transpiration (Yang et al., 2023). However, estimation of ET via remote sensing remains a
significant challenge with implications for understanding of vegetation response to drought, fire
risk, and the accounting of freshwater resources.

One challenge for ET remote sensing is that, unlike some surface properties such as
temperature, we are unable to directly sense the flux of water or latent heat associated with ET
electromagnetically. Therefore, ET products must leverage modelling approaches - either
physical, hybrid, or machine learning - constrained by the data that *is* observable via remote
sensing. These modelling approaches for ET often assume - implicitly or explicitly - the response
of evaporation and transpiration to environmental drivers, such as drought or variations in land
cover.

Alternatively, surface flux equilibrium (SFE) is a data-driven method for estimating ET
directly from atmospheric conditions without relying on soil or vegetation parameterization.
The concept of surface flux equilibrium was first proposed by McColl et al. (2019) and states
that, under many circumstances, the atmosphere and land surface are coupled so that changes
in surface fluxes (including ET) are reflected in atmospheric temperature and humidity. This
approach has several advantages over other ET estimation methods. It requires no information
about vegetation, soil, or subsurface properties. It also makes no assumptions about root-zone
moisture status or vegetation response to water availability. This means it is well suited for
hydrological research attempting to interrogate the relationship between ET and water
availability or between ET and vegetation cover (or other biogeographic drivers). Additionally,
SFE includes no tunable parameters and can be computed easily using only three inputs - air
temperature, humidity, and net radiation - each of which is readily available at global scales
(McColl and Rigden, 2020).

However, more complex ET estimation methods would be expected to outperform SFE
in many settings due to its extreme simplicity and lack of adjustable parameters. Nevertheless,
previous SFE implementation and validation efforts indicate that SFE performance is
comparable - or even better than - other ET estimation methods at the point- and watershed-
scale (Chen et al., 2021; McColl and Rigden, 2020; Thakur et al., 2025). For example, SFE ET has
been found to be within the range of in situ measurement errors at a selection of inland eddy
covariance towers, an upper limit on the performance of any ET estimate (McColl and Rigden,
2020). Thakur et al. (2025) also calculated SFE ET at inland eddy covariance sites across the
continental United States (CONUS) using tower-based temperature, humidity, and net
radiation. They found that SFE ET outperformed remotely sensed ET from MODIS (Mu et al.,
2011) as well as from three ET algorithms using data from the ECOsystem Spaceborne Thermal





Radiometer Experiment on Space Station (ECOSTRESS): the Simplified Surface Energy Balance
(Savoca et al., 2013) SSEBop, (Savoca et al., 2013), the atmosphere-land exchange inverse
disaggregation algorithm (DisALEXI) and the Priestley-Taylor Jet Propulsion Laboratory model
(PT-JPL, Fisher et al., 2020).

Thakur et al. (2025) further investigated the impact of input data on SFE performance by

calculating SFE ET using three scenarios: only eddy covariance data, by using the North
American Land Data Assimilation System (NLDAS, Xia et al., 2012) for temperature and humidity
and the Clouds and the Earth's Radiant Energy System instrument (CERES, Doelling et al., 2013)
for net radiation, and by finally using NLDAS for temperature and humidity and MODIS for net
radiation. All three SFE ET implementations compared favorably to tower-based ET with $R^2$ of
0.70, 0.68, and 0.67 for the tower-based SFE, CERES-based SFE, and MODIS-based SFE,
respectively. This suggests that the emergent simplicity of ET that SFE takes advantage of is
robust to choices of input data, at least at the scale of eddy covariance towers.

The only gridded estimates of SFE ET are reported by Chen et al. (2021), who calculated

monthly ET at 0.125° across CONUS using net radiation from CERES and 2-m temperature and
humidity from North American Regional Reanalysis (NARR, Mesinger et al., 2006). They
compared SFE ET to estimates from the Coupled Model Intercomparison Project phase 6
(CMIP6, Eyring et al., 2016) and to water balance-based ET estimates available at large
catchments across CONUS. The error in the water balance-based estimates provides a minimum
possible error, below which ET estimation approaches cannot be distinguished due to errors in
the underlying reference data. They found that SFE ET errors are comparable to the error of the
catchment water balances and that SFE outperforms the reanalysis (NARR) and most CMIP6
models.

However, even this sole gridded implementation of SFE - while promising - is unable to

provide a thorough evaluation of the SFE approach because the comparison datasets each have
their own unquantified uncertainties. Therefore, disagreement between SFE and CMIP6 cannot
be attributed to either dataset because their errors cannot be distinguished. One solution to
this is the statistical evaluation approach of triple collocation. Using triple collocation and its
updated counterpart, extended triple collocation (McColl et al., 2014), it is possible to compare
three datasets with co-located measurements and estimate two important performance
metrics: (1) the variability in the random error of each dataset and (2) the correlation between
the measured value and the underlying 'true' variable. Both performance metrics can be
calculated without reference to this unknowable 'true' variable, in this case ET, and without
assuming the error of any of the three comparison datasets.





Triple collocation - sometimes also referred to as the 'three-cornered hat' approach -
has been widely used in evaluating datasets where a 'truth' or reference dataset is unavailable,
for example in the evaluation of datasets for soil moisture (Draper et al., 2013; Gruber et al.,
2016; Scipal et al., 2008), ocean winds (Caires & Sterl, 2003), precipitation (Alemohammad et al,
2015, Burnett et al 2020), sensible heat and carbon fluxes (Alemohammad et al, 2017), ET
(Khan et al., 2018), near-surface air temperature and specific humidity (Sun et al., 2021), and
terrestrial water storage (Ferreira et al., 2016). It can also be used to estimate the coupling of
multiple variables, for example latent heat and soil moisture (Crow et al., 2015). Given three
datasets with observations of the same state variable, each with their own non-correlated
random errors, comparison of the three datasets via triple collocation enables calculation of
each dataset's random error variance (Stoffelen, 1998).
Here, we accomplish two steps in advancing the estimation of ET. First, we release the
first publicly available, gridded dataset of daily SFE ET. We calculate this dataset at 4 km
resolution across the continental United States (CONUS) using gridMET for 2-m temperature
and humidity and net radiation from ERA5-Land. Second, we compare our gridded estimates of
SFE ET to three other remotely sensed ET estimates: Global Land Evaporation Amsterdam
Model Version 4 (GLEAM, Miralles et al., 2011), FluxCom (Jung et al., 2019), and ERA5-Land
(Muñoz-Sabater et al., 2021). In addition to comparing the spatial pattern and variance of all
datasets, we further use the statistical method of extended triple collocation following McColl
et al. (2014) to calculate the error statistics of each dataset, despite lacking observations of
'true' ET (Gruber et al., 2016; McColl et al., 2014; Stoffelen, 1998).

## 2. Methods

### 2.1. Calculating ET from atmospheric conditions assuming surface flux equilibrium
We calculate daily ET after McColl et al. (2019) by assuming that the near-surface atmosphere is
in a state of 'surface flux equilibrium' where atmospheric conditions at the boundary layer
reflect the recent fluxes of latent ($\lambda E$) and sensible (H) heat on the Earth's surface. If this is the
case, then increasing ET (i.e. increasing latent heat) will correspond with diminished sensible
heat and result in both atmospheric cooling and increased humidity. The ratio of sensible and
latent heat fluxes - known as the Bowen ratio (B) - can therefore be approximated by
temperature and humidity at the boundary layer, so long as atmospheric conditions reflect the
integrated signal of fluxes on the Earth's surface.

We use 2-m air temperature ($T_a$) and relative humidity ($q_a$) from gridMET (Abatzoglou,
2013) to estimate the Bowen ratio, where $R_v$ = 461.5 (J kg$^{-1}$ K$^{-1}$) is the gas constant for water



vapor, $C_p$ = 1005 (J kg$^{-1}$ K$^{-1}$) is the specific heat capacity of air at constant pressure, and $\lambda$ = 2.56
x 10$^6$ (J kg$^{-1}$) is the latent heat of vaporization of water (Eq 1).

$$B = \frac{H}{\lambda E} \approx \frac{R_v c_p T_a^2}{\lambda^2 q_a}$$  Eq. 1

We choose gridMET due to its relatively fine spatial resolution of 4 km and its availability
at the daily timescale across CONUS. Net radiation ($R_n$) allows conversion from the Bowen ratio
to ET (Eq 2). We use $R_n$ from ERA5-Land (Muñoz-Sabater et al., 2021) because of its high
agreement with in situ measurements across CONUS (Yin et al., 2023). Additionally, we assume
a ground heat flux (G) of 10%. We do not evaluate SFE ET on any days with negative $R_n$.

$$\lambda E = (1 + B)^{-1}(R_n - G)$$  Eq. 2


### 2.2 Triple collocation error estimation

Triple collocation assumes a linear error model for each dataset, where the observed value for
a given dataset ($x_i$) is assumed to be a linear function of the "true" ET ($T$) obscured by a
constant additive bias ($\alpha$), a constant multiplicative bias ($\beta$) and a time-varying additive random
error with zero mean ($\epsilon$) (Eq 3).

$$x_i = \alpha_i + \beta_i T + \epsilon_i$$  Eq. 3


In addition to assuming a linear error model for each dataset, triple collocation further
assumes that the errors of each dataset are stationary and uncorrelated both with each other
and with the unknown truth (Gruber et al., 2016; McColl et al., 2014).
With these assumptions, the variance of each dataset ($Q_{11}$, $Q_{22}$, and $Q_{33}$) represents the
sensitivity of the dataset to variations in the true signal (via the product of $\beta_i$ and $\sigma_T$) plus the
variance of the random error ($\sigma_{\epsilon_i}^2$) (Eq 4).

$$Q_{ii} = \sigma_i^2 = \beta_i^2 \sigma_T^2 + \sigma_{\epsilon i}^2$$  Eq. 4

Covariance between pairs of datasets (e.g. $Q_{12}$, $Q_{13}$, and $Q_{23}$) likewise provides
information about each dataset's sensitivity to the true unknown ET via $\beta_i$ and $\sigma_T$. (Eq 5).

$$Q_{ij} = \sigma_{ij}^2 = \beta_i \beta_j \sigma_T^2$$  Eq. 5



By treating the product of $\beta_i$ and $\sigma_T$ as a single unknown variable, the equations for the
variance and covariance of each dataset and dataset pair result in six equations and six
unknowns. These can be solved to calculate the standard deviation of the random error of each
dataset, $\sigma_\varepsilon$ (Eq 6).

$$
\sigma_\varepsilon = \begin{bmatrix} \sqrt{Q_{11} - \dfrac{Q_{12}Q_{13}}{Q_{23}}} \\ \sqrt{Q_{22} - \dfrac{Q_{12}Q_{23}}{Q_{13}}} \\ \sqrt{Q_{33} - \dfrac{Q_{13}Q_{23}}{Q_{12}}} \end{bmatrix}
$$

Eq. 6

The absolute values of $\beta_i$ cannot be separated from the absolute value of $\sigma_T$. However,
many studies assume $\beta_i = 1$ for one dataset - effectively choosing it as a reference dataset
which has no multiplicative bias - and calculate $\beta_i$ for the other two datasets *relative* to the
actual unknown multiplicative bias of the reference dataset. In this study, however, we do not
separate $\beta_i$ and $\sigma_T$.
Extended triple collocation further allows the calculation of the correlation between
each dataset and the unknown truth, $R_T$, while requiring no additional information (McColl et
al., 2014); Eq 7).

$$
R_{T,i}^2 = \begin{bmatrix} \dfrac{Q_{12}Q_{13}}{Q_{11}Q_{23}} \\[2ex] \dfrac{Q_{12}Q_{23}}{Q_{22}Q_{13}} \\[2ex] \dfrac{Q_{13}Q_{23}}{Q_{33}Q_{12}} \end{bmatrix}
$$

Eq. 7

Triple collocation requires several assumptions, all of which are likely to be at least
partially violated (e.g., Yilmaz and Crow, 2014). However, these assumptions are not unique to
triple collocation. Gruber et al. (2016) showed that more common validation strategies
implicitly require the same assumptions. For example, if we were to instead estimate the
correlation coefficient and root-mean-squared error (RMSE) between SFE ET and another
reference ET product, we would be implicitly making the same assumptions.
***2.3. Comparison ET datasets***
We compare SFE ET to ET from FluxCom, GLEAM version 4, and ERA5-Land. We compare all ET
datasets over the years 1980 to 2016, which represents the maximum overlap in temporal
coverage between all four datasets. Additionally, we resample each dataset to match the native





resolution of FluxCom at 0.5°. We match the FluxCom resolution because it is the coarsest. We
choose to compare SFE to these particular three ET datasets not just because they are
commonly used, but also to minimize violation of the triple collocation assumptions,
particularly the assumption of independent errors between datasets. This is commonly
achieved by using datasets that differ in their input data sources and modeling frameworks
(Gruber et al., 2016; McColl et al., 2014). We also remove the seasonal cycle from each dataset
by subtracting the 30-day rolling average from each day (Chen et al., 2018; Draper et al., 2013;
Miralles et al., 2010). This ensures that differences in the seasonality and timing of ET do not
impact the triple collocation analysis and has been shown to improve error estimation with
triple collocation for ET datasets specifically (He et al., 2023). Finally, we use extended triple
collocation to calculate the standard deviation of the random error and the correlation
coefficient of each dataset (see Sec 2.2 above). Because we have four comparison datasets and
triple collocation requires just three, we are able to repeat our estimates of each dataset's
error statistics once for each possible 'triplet' (i.e. combination) of three datasets. Convergence
of the error estimates regardless of the triplet chosen increases the robustness of the triple
collocation assumptions and improves confidence in our calculated values (Draper et al., 2013;
He et al., 2023). In addition to performing triple collocation, we also compare the four datasets
via a general analysis of the variance and spatial patterns of ET.

The FluxCom dataset we choose for our triple collocation analysis uses machine learning

to upscale eddy covariance measurements from flux towers based on satellite and
meteorological inputs. FluxCom provides an ensemble of latent heat estimates trained using
different meteorological datasets. In order to have the longest data record with daily
resolution, here we use the single FluxCom ensemble member trained with the CRUNCEPv6
reanalysis product (Wei et al., 2014), as opposed to the mean of all possible FluxCom ensemble
members. However, the different model setups (each with a different weather model) were
previously found to have similar performance (Jung et al., 2019). In addition to the climate data
from CRUNCEP, FluxCom uses radiation data from CERES (Doelling et al., 2013), precipitation
from the Global Precipitation Climatology Project (GPCP, Huffman et al., 2001), and
temperature, land cover, and other reflectance indicators from MODIS. The FluxCom model is
run per plant functional type and then combined into a single estimate by weighting each plant
functional type's fractional areal coverage of the pixel (Jung et al., 2019).

GLEAM estimates ET by using remote sensing and reanalysis data to force a hybrid

model which includes modules for canopy interception, potential evapotranspiration, soil water
content, and vegetation response to evaporative stress. Although FluxCom and GLEAM have
some remote sensing inputs in common, for example radiation from CERES and vegetation
information from MODIS, Gleam Version 4 takes a hybrid modelling approach and does not rely
fully on machine learning like FluxCom. Specifically, GLEAM version 4 primarily uses physical





modelling modules with only a single module – for evaporative stress – using a deep neural
network trained using in situ data from eddy covariance towers and sap flow measurements
(Koppa et al., 2022; Martens et al., 2017; Miralles et al., 2025).  This is in contrast to GLEAM
version 3, which estimates evaporative stress empirically as a function of soil moisture and
vegetation optical depth - both from microwave remote sensing inputs. Additionally, GLEAM
Version 4 calculates ET using Penman's equation (as opposed to Priestley-Taylor, used in
Version 3) and also updates the multi-layer water balance model so that vegetation access to
groundwater can be represented. However, in GLEAM Version 4, plant rooting depths are static
for each land cover within the groundwater scheme and there is still a prescribed multiplicative
stress function to determine how vegetation responds to soil moisture stress. GLEAM is the
only dataset in our comparison set which partitions ET between evaporation, transpiration, and
interception. We use the variable referring to the total evaporation (E) to best match the other
ET estimates.

Finally, ERA5-Land uses the near-surface atmospheric reanalysis from ERA5, which
assimilates observations from a range of satellites and in situ observation networks for many
variables including land surface temperature, precipitation, wind speed, and soil moisture
(Hersbach et al., 2020). ERA5-Land then takes the atmospheric states from ERA5 and re-runs
the land surface model component at a finer resolution (9 km) offline (Muñoz-Sabater et al.,
2021). This allows for additional and refined land surface parameterizations and corrections.
Unlike FluxCom and GLEAM, ERA5-Land has no machine learning components. For our analysis,
we sum the hourly latent heat flux output of ERA5-Land to daily totals and then resample
bilinearly to match the coarser 0.5° FluxCom grid. Finally, both ERA5-Land and FluxCom report
latent heat flux in units of energy per unit area, which we convert to ET (mm/day) by dividing by
the latent heat of vaporization ($\lambda$ = 2.56 × 10⁶ J kg⁻¹).

### 2.4. Comparing performance across biogeographical factors

We compare the resulting $\sigma_\varepsilon$ and $R_T$ estimates from triple collocation across a variety of
biogeographical factors - specifically climate, elevation, land cover type, and the distance to the
coast - to better understand under what conditions SFE ET performs well and how its
performance across biogeography compares to that of the other ET estimates.

We calculate the mean annual precipitation at each pixel using monthly precipitation (P)
from 1991 to 2020 from TerraClimate (Abatzoglou, 2013).We use elevation from MERIT Hydro
(Version 1.0.1., (Yamazaki et al., 2019). For land cover, we use the National Land Cover
Database (NLCD) land cover map from 2021 (Dewitz, 2024). We consider the land cover types
of forest (combining deciduous, evergreen, and mixed forests), shrub, grassland, wetland
(combining woody and herbaceous wetlands), and agricultural (cultivated crops).



We further analyze the performance of each dataset by each pixel's distance from the
coast because the assumptions of SFE are likely to be violated near the ocean (McColl et al.,
2019). This is because in coastal regions, ocean moisture and temperature are expected to be a
strong control on land surface fluxes. We calculate the distance of each pixel centroid from the
nearest coast using the TIGER/Line Coastline National Shapefile (United States Census Bureau,
2019). We also exclude pixels from all analyses if their centroid overlaps with the ten largest
water bodies in CONUS (ArcGIS Data and Maps, 2023).

**3. Results**
***3.1. Surface flux equilibrium ET across CONUS from 1979 to 2024***
Here, we publicly release a dataset of daily SFE ET from 1979 to 2024 at 4 km resolution across
CONUS (see Data Availability section). The spatial mean (shown in Figure 1a) follows expected
patterns across CONUS - with an aridity driven gradient from West to East and a radiation
driven gradient from North to South in the Eastern US. The temporal variability in daily ET
calculated using the SFE approach is consistent with the comparison datasets (Figure S1).
However, SFE has a larger standard deviation across much of CONUS - particularly the Western
US - than FluxCom and GLEAM. Across several sample pixels, chosen as heavily vegetated
examples spanning multiple regions, the seasonal cycle of mean annual ET is likewise
comparable across all four ET estimates, although the timing of maximum summer ET each year
varies between datasets (Figure 1b-g).
Although the magnitude of mean annual continental ET is most similar between SFE and
FluxCom (Figure 2), the pattern of interannual variability which matches SFE the best is that of
GLEAM ($\rho$= 0.56). The two datasets with the overall closest match in ET interannual variability,
however, are FluxCom and ERA5-Land ($\rho$= 0.71). Although SFE and FluxCom each have
intermediate magnitudes of mean continental ET relative to GLEAM and ERA5, both datasets -
and FluxCom in particular - also have the lowest interannual variability magnitude (8 mm/year
standard deviation for FluxCom and 12 mm/year for SFE, compared to 22 and 28 mm/year for
ERA5-Land and GLEAM, respectively). Across the entire average record, the mean annual ET
from SFE (598 mm/yr) sits roughly in the center of the four datasets, with GLEAM the lowest
(555 mm/yr) and ERA5-Land the highest (641 mm/yr).



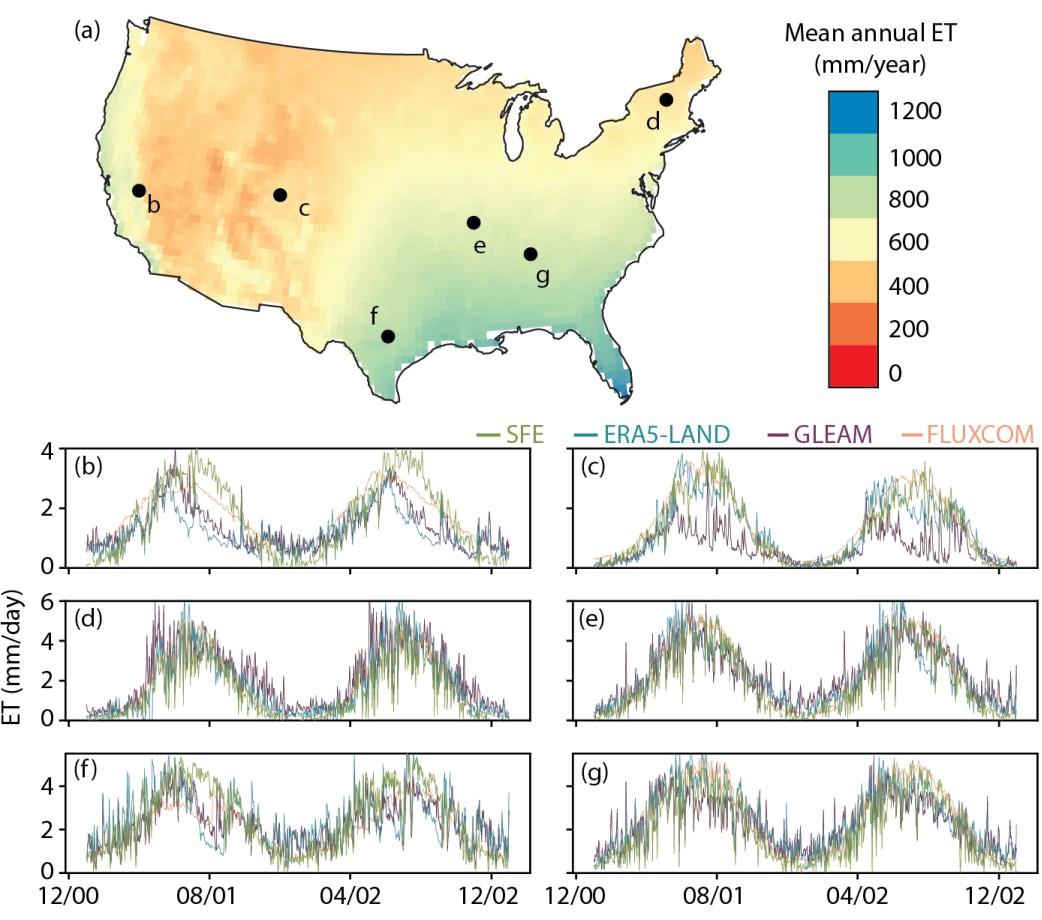

*Figure 1. Mean annual SFE ET across CONUS from 1979 to 2024. Points show timeseries for example pixels for SFE (green), ERA5-Land (blue), GLEAM (purple) and FluxCom (pink).*





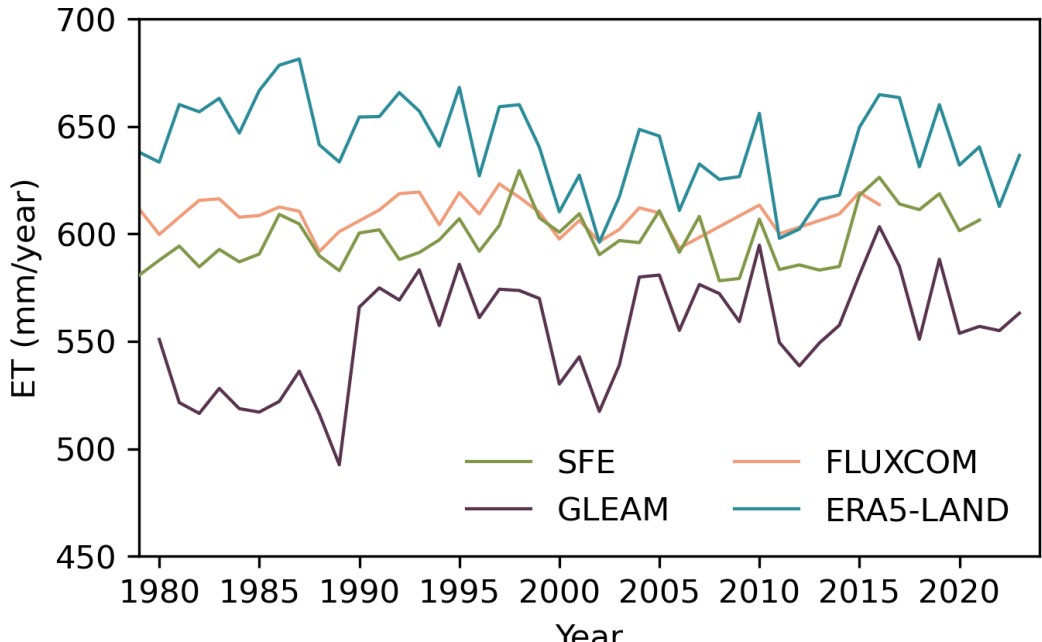


*Figure 2. Interannual variability in mean annual ET across CONUS from 1979 through the record*
*length of each dataset.*





### 3.2. SFE is the only dataset that performs well in terms of both the standard deviation of the random error and the correlation coefficient



SFE performance as estimated by triple collocation is comparable - and even exceeds - the
performance of the comparison datasets across much of CONUS, despite its extreme simplicity,
lack of tunable parameters, and relatively small number of assumptions (Figure 3). SFE,
FluxCom, and GLEAM show a strong divide in performance between the Western and Eastern
US. SFE and FluxCom both have the lowest $\sigma_\varepsilon$ and highest $R_T$ in the Western US compared to
the Eastern US. In contrast, GLEAM has lower $\sigma_\varepsilon$ in the Western US, but higher $R_T$ in the Eastern
US. ERA5-Land shows more heterogeneity in performance across space - especially compared
to SFE and FluxCom - and has no clear performance gradient between the Western and Eastern
US.



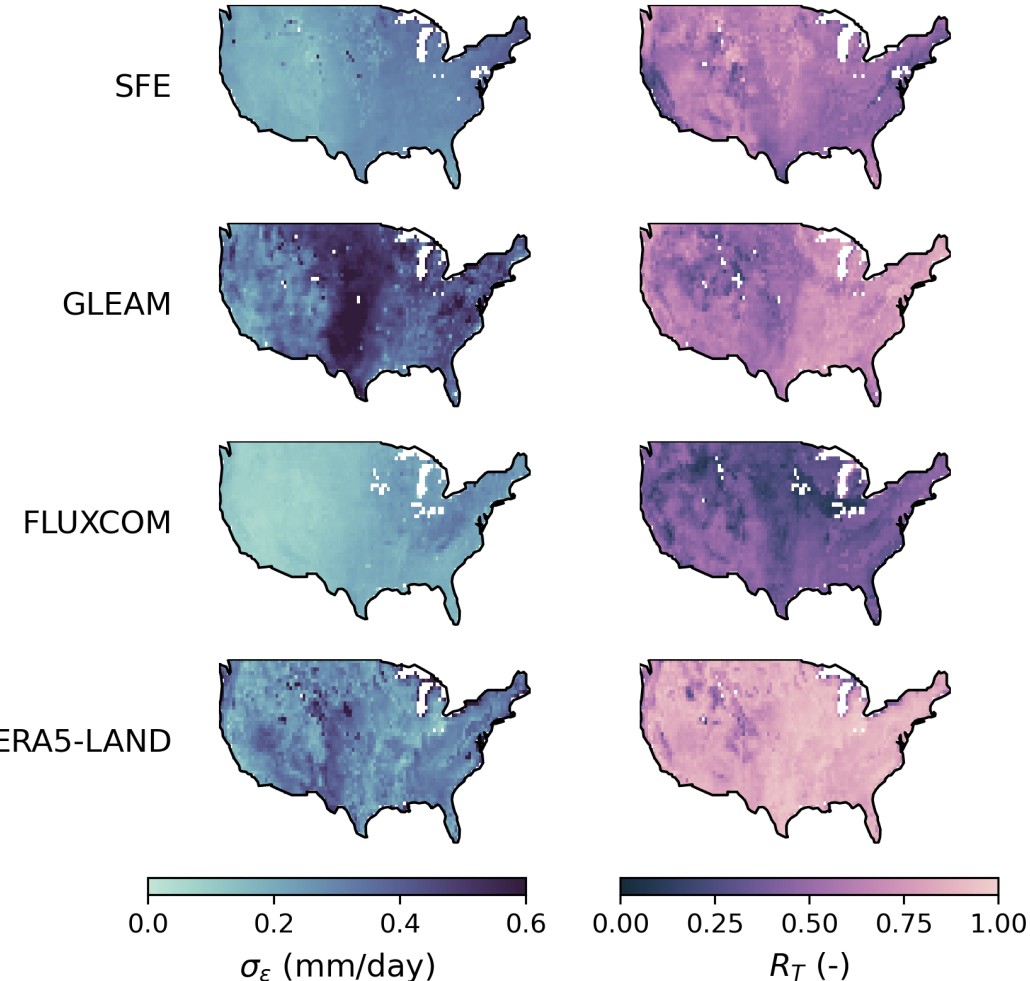


Figure 3. The standard deviation of the random error, $\sigma_\varepsilon$ (left) and correlation coefficient to the truth, $R_T$ (right) for each dataset averaged across all triplet combinations. Increasingly light colors are better performance. White pixels have no valid data for any triplet.


Despite its simplicity, SFE is the best or second-best dataset according to both $\sigma_\varepsilon$ and $R_T$ across more than half of CONUS (Figure 4). SFE has the lowest or second lowest $\sigma_\varepsilon$ and highest or second highest $R_T$ across 65.8% and 45.7% of pixels across CONUS, respectively (Figure 4, Table 1), mostly in the Western US.

SFE's high performance with regards to both $\sigma_\varepsilon$ and $R_T$ is unique among the comparison datasets. Other than SFE, the datasets with the best $\sigma_\varepsilon$ and $R_T$, respectively, have the lowest



performance for the complementary metric. For example, FluxCom has the lowest $\sigma_\varepsilon$ across the
majority of CONUS, but it also has the lowest $R_T$ (Figure 4). The opposite is true for ERA5, which
is the highest performing dataset according to $R_T$ across much of CONUS but frequently has the
worst performance according to $\sigma_\varepsilon$, particularly in the US Southwest. SFE is the only dataset
which consistently has high performance according to both metrics.


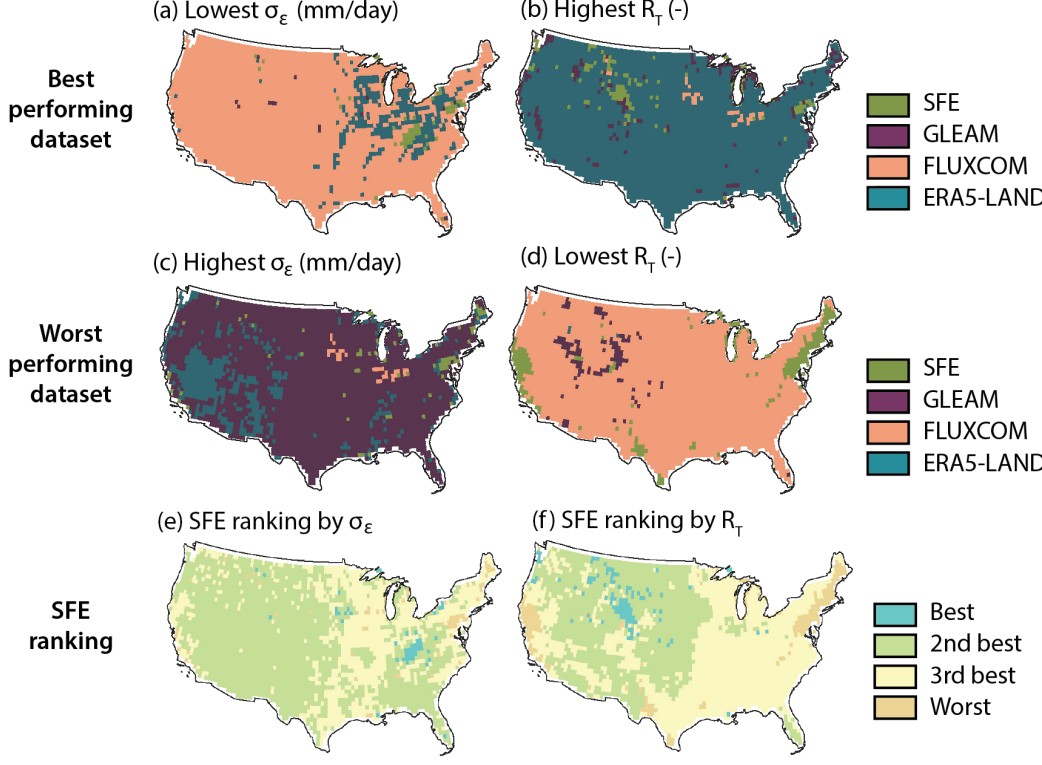



*Figure 4. Summary of relative performance of all four datasets. The dataset with highest*
*performance for the standard deviation of the random error, $\sigma_\varepsilon$ (a) and the correlation*
*coefficient with 'true' ET, $R_T$ (b) for each pixel. The worst performing datasets for $\sigma_\varepsilon$ (c) and $R_T$*
*(d). The relative ranking of SFE for $\sigma_\varepsilon$ (e) and $R_T$ (f). The total number of pixels (and relative*
*percent of pixels) of each color are shown in Table 1. Pixels with centroids within 4 km (i.e., one*
*pixel) of the border have been removed.*



*Table 1. (Top) The number of pixels where each dataset has the best performance according to the standard deviation of the random error, $\sigma_\varepsilon$, and the correlation coefficient to the truth, $R_T$. (Bottom) The number of pixels by SFE ET ranking.*

| | By $\sigma_\varepsilon$ | | By $R_T$ | |
|---|---|---|---|---|
| **Best dataset** | | | | |
| | Pixels | Percent | Pixels | Percent |
| SFE | 58 | (1.9%) | 117 | (3.9%) |
| GLEAM | 17 | (0.6%) | 161 | (5.3%) |
| FLUXCOM | 2665 | (87.9%) | 34 | (1.1%) |
| ERA5-Land | 292 | (9.6%) | 2720 | (89.7%) |
| **Ranking of SFE** | | | | |
| | By $\sigma_\varepsilon$ | | By $R_T$ | |
| | Pixels | Percent | Pixels | Percent |
| 1st | 48 | (1.6%) | 105 | (3.5%) |
| 2nd | 1946 | (64.2%) | 1279 | (42.2%) |
| 3rd | 975 | (32.2%) | 1455 | (48.0%) |
| 4th | 63 | (2.1%) | 193 | (6.4%) |

We note that the estimates of $\sigma_\varepsilon$ and $R_T$ are consistent between triplets, indicating $\sigma_\varepsilon$ and $R_T$ estimates are robust to the choice of comparison datasets (Figure 5). Individual $\sigma_\varepsilon$ and $R_T$ maps for each dataset and triplet combination are shown in Figures S2 and S3. However, not all pixels have valid results for each triplet combination, which occurs when either $\sigma_\varepsilon$ is negative for one or more of the datasets or if any $R_T$ are greater than one. Figure 6 shows the total number of triplets which are valid for each pixel. The triplets with the most invalid pixels are those where FluxCom and ERA5-Land are both included. Invalid pixels are also more common in the Eastern US rather than the Western US. Even in the East, however, SFE - our main estimate of interest - still has at least one valid triplet in 96% of pixels and at least two valid triplets in 86% of pixels. SFE has three valid triplets - the maximum possible number for our four dataset analysis - in 55% of pixels.



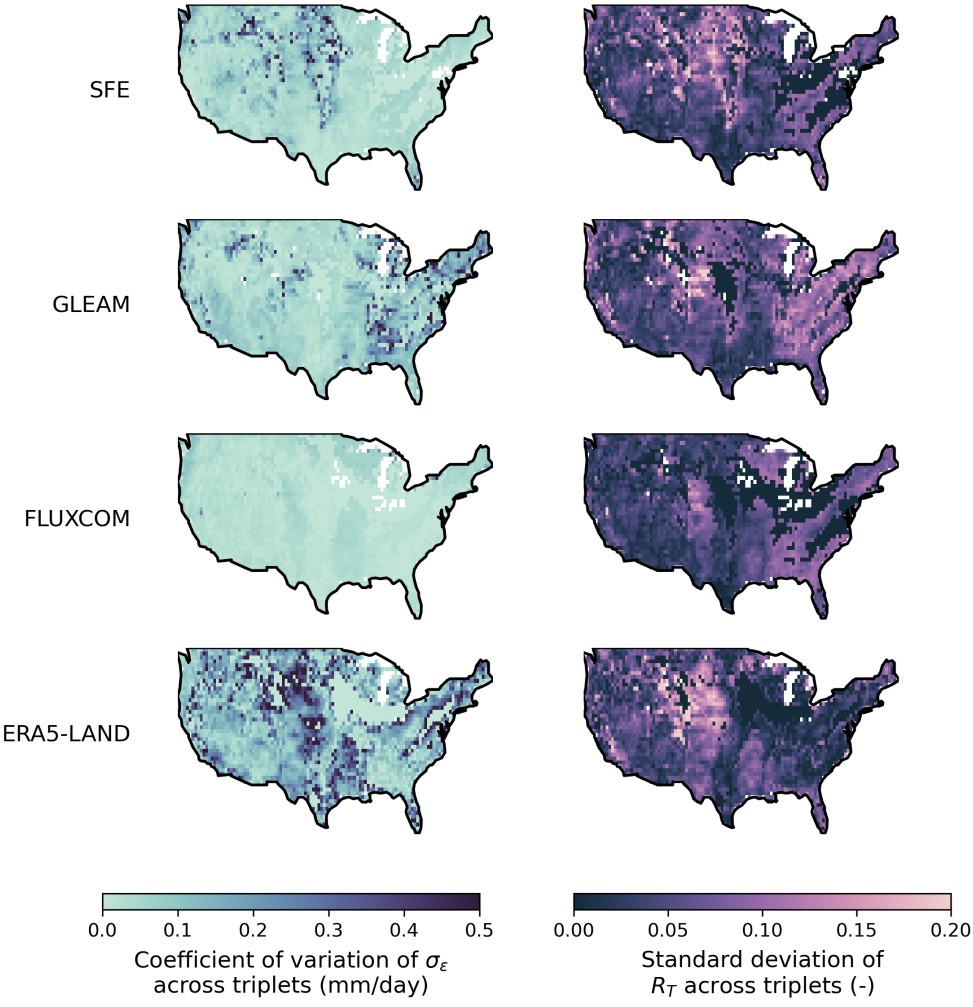

*Figure 5. (left) The coefficient of variation of $\sigma_\varepsilon$ for each dataset across all possible triplet combinations with valid data. White pixels have no valid data for any triplet. (right) The standard deviation of $R_T$ for each dataset across all possible triplet combinations with valid data. White pixels have no valid data for any triplet and black pixels have only one triplet combination with valid data.*





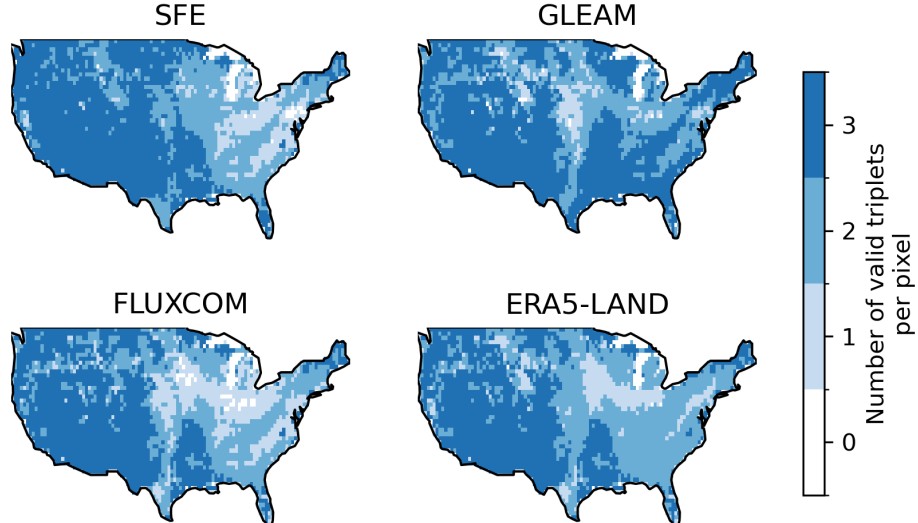

*Figure 6. The total number of triple collocation estimates - one from each possible combination*
*of datasets - that are averaged for each pixel and dataset combination. Pixels with no valid*
*triple collocation results for any triplet are shown in white. The maximum number of valid*
*triplets is three.*

### 3.3. Performance across biogeographical factors

Comparing the trends of $\sigma_\varepsilon$ (Figure 7) and $R_T$ (Figure 8) across mean annual precipitation, elevation, landcover, and the distance to large water bodies shows that SFE performance is not more sensitive to any of these biogeographical factors than the comparison datasets. Even when comparing SFE performance with coastal proximity - a factor where we expect to see performance degradation due to the violation of SFE assumptions (McColl and Rigden, 2020) - the coastal proximity penalty of SFE is comparable to that of ERA5-Land. Indeed, ERA5-Land shows the sharpest decrease in performance within 20 km of the coast out of any of the datasets, however both SFE and ERA5-Land continue to show improved performance even up to 120 km inland. Neither GLEAM nor FluxCom have a strong relationship between coastal proximity and performance.

Likely due to its correlation with coastal proximity, SFE also has decreased performance at lower elevations with respect to both evaluation metrics. FluxCom and GLEAM likewise show their highest $\sigma_\varepsilon$ at low elevations relative to higher elevations, with FluxCom $\sigma_\varepsilon$ peaking around 500 m a.s.l. and GLEAM $\sigma_\varepsilon$ around 1000 m. a.s.l. All three datasets continue to have decreased $\sigma_\varepsilon$ as elevation increases. The relationship between elevation and $R_T$ is relatively flat for SFE and FluxCom in the intermediate elevations, with the lowest $R_T$ at the extreme low and high



elevations. GLEAM and ERA5, however, have continuously decreasing $R_T$ with increasing
elevation, and the lowest $R_T$ at elevations exceeding 2000 m a.s.l.

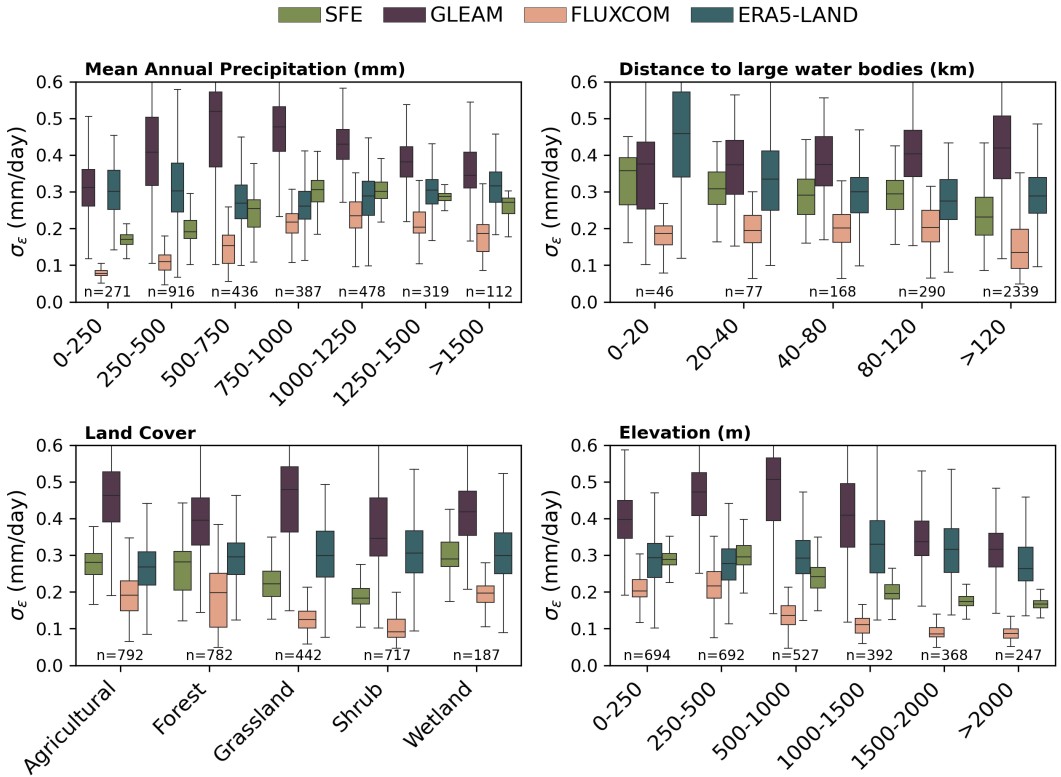

*Figure 7. The standard deviation of the random error, $\sigma_\varepsilon$, for each ET dataset across mean*
*annual precipitation, the distance to large water bodies, elevation, and land cover. The number*
*of pixels in each category per ET dataset is shown below boxes.*

The $\sigma_\varepsilon$ for SFE, GLEAM, and FluxCom is lowest at the driest and wettest pixels and
highest at pixels with intermediate precipitation. However, the $\sigma_\varepsilon$ for GLEAM peaks at the 500-
750 mm/year bin whereas FluxCom and SFE have the highest $\sigma_\varepsilon$ at slightly wetter locations,
receiving between 1000-1250 mm/year. ERA5-Land, on the other hand, has a weaker
relationship between MAP and $\sigma_\varepsilon$. ERA5-Land has the opposite pattern than the other datasets
and shows the highest $\sigma_\varepsilon$ at the driest and wettest pixels with lower $\sigma_\varepsilon$ at intermediate aridity.
The relationship between MAP and $R_T$ follows that of MAP and $\sigma_\varepsilon$ in general, however $R_T$ does



not increase at the wettest pixels to the same degree as for the $\sigma_\varepsilon$. For example, SFE has
continually decreasing $R_T$ as MAP increases with only a minimal increase in performance at the
pixels with >1500 mm/year of precipitation.

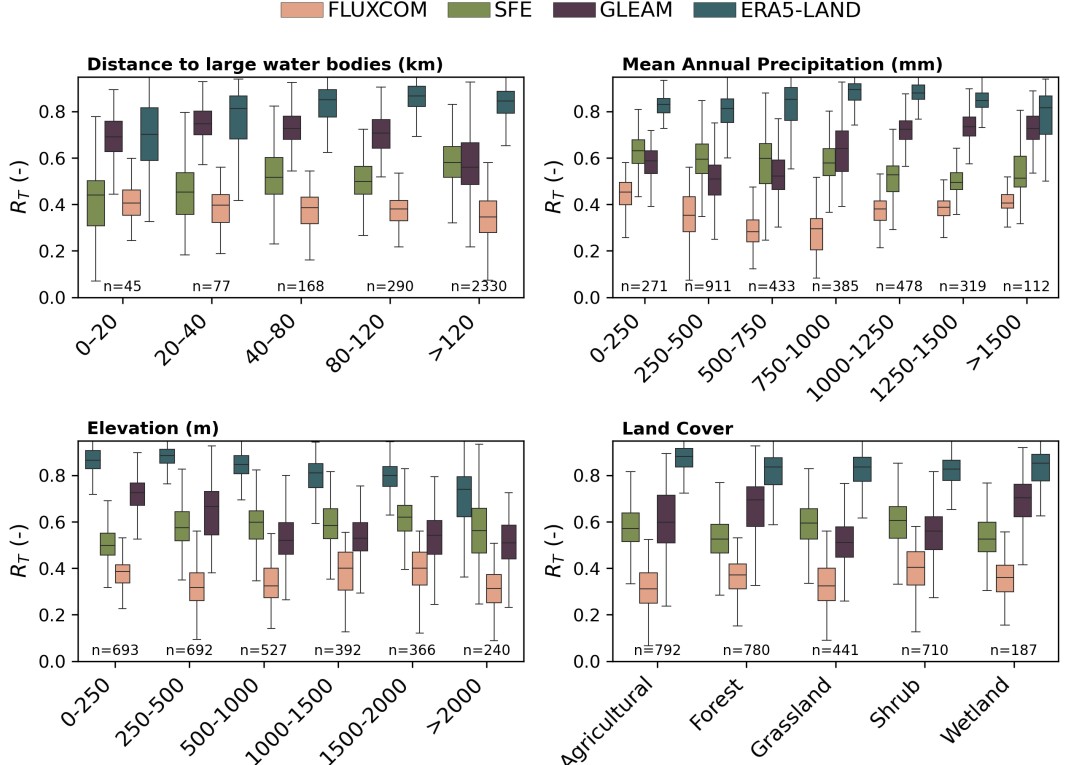


*Figure 8. The correlation coefficient, $R_T$, for each ET dataset across mean annual precipitation,*
*the distance to large water bodies, elevation, and land cover. The number of pixels in each*
*category per ET dataset is shown below boxes.*

The performance variability across land cover is not consistent between any of the
datasets. ERA5-Land has the lowest $\sigma_\varepsilon$ and highest $R_T$ in agricultural pixels, GLEAM in forest
pixels, and FluxCom in shrubland pixels. The SFE $R_T$ is similar across all land cover types but SFE
$\sigma_\varepsilon$ is highest in wetlands, followed by forest and agricultural pixels. Forested pixels also have a
greater spread in $\sigma_\varepsilon$ for FluxCom and SFE compared to the other land cover types. SFE $\sigma_\varepsilon$ is
lowest in shrublands, followed by grasslands. FluxCom $\sigma_\varepsilon$ is likewise lowest for grassland and



shrublands, which is the opposite of ERA5-Land, with the highest $\sigma_\varepsilon$ in grasslands and
shrublands.

**4. Discussion**
***4.1. Which ET estimate is most accurate?***
While triple collocation reveals that SFE is rarely the highest performing dataset, it is the
second-best performing dataset across much of CONUS for both $\sigma_\varepsilon$ and $R_T$ (Figure 4e,f). In
addition, we find that datasets which outperform SFE only exhibit better performance for one -
not both - of either $\sigma_\varepsilon$ and $R_T$. That SFE performs well - although not the best - for both metrics
suggests its usefulness for a variety of applications, particularly those where it is not clear *a*
*priori* whether having high $R_T$ or low $\sigma_\varepsilon$ is most useful. Furthermore, SFE may be a particularly
good choice for studies interested in the response of ET to water limitations. Unlike the
explicitly assumed dependence of ET on hydrologic conditions in ERA5-Land or the implicitly
assumed dependence of GLEAM and FluxCom (which is limited by the constraints of the
machine learning structure and input data), SFE contains no *a priori* assumptions about the
effect of water stress on ET. Our release, alongside this manuscript, of a daily, 4km resolution
CONUS-wide dataset of SFE-based ET spanning 1979 to 2024 should facilitate future
applications of SFE for scientific analyses.

SFE is generally the second-best dataset regardless of metric, while alternative datasets
with low random noise also have low correlation with the truth and vice versa. For example,
across the four datasets tested, FluxCom has the lowest (most desirable) $\sigma_\varepsilon$ across the majority
of CONUS pixels (Figure 4a). However, it also has the lowest (least desirable) $R_T$ more often than
any other datasets (Figure 4d). ERA5-Land shows the converse relationship, with the highest
(most desirable) $R_T$ in almost all pixels compared to all other datasets, but poorer relative
performance with regard to $\sigma_\varepsilon$ (Figure 4b,c). How is this possible? To understand why, note that
the triple collocation error model implies that,

$$R_{T,i}^2 = \frac{\beta_i^2 \sigma_T^2}{\beta_i^2 \sigma_T^2 + \sigma_{\epsilon,i}^2}$$    Eq. 8

as shown in McColl et al. (2014). For a dataset to exhibit both the lowest $R_T$ and lowest $\sigma_\varepsilon$
requires that $\beta$ is also sufficiently small ($\sigma_T$ is the same for each dataset and does not impact
the ranking). An extreme example would be a dataset that simply set ET to a fixed
climatological value and exhibited no temporal variability, for which $\beta = 0$ and $R_T = 0$, even
when $\sigma_\varepsilon$ is small. At the other extreme, for a dataset to exhibit both highest $R_T$ and highest $\sigma_\varepsilon$
requires $\beta$ to be sufficiently large. In the limit of $\beta \to \infty$, $R_T = 1$, even when $\sigma_\varepsilon$ is large. The
relative importance of choosing a dataset with a low $\sigma_\varepsilon$, a high $R_T$, or a low bias (which is not



assessed here), depends on the application for which the ET dataset will be used (Entekhabi et
al., 2010).
Beyond choosing a single dataset for a particular application, it is also possible to
average multiple ET estimates into a single dataset weighted by each dataset's performance.
While not often practical for large-scale use, He et al., (2023) used triple collocation to estimate
an 'optimal' ET product over China by weighting each dataset by its uncertainty. Burnett et al
(2020) also used this approach to generate a new rainfall product for the Congo River Basin.
Such an approach was also proposed as a possible way forward by the WAter Cycle Multi-
mission Observation Strategy (WACMOS) project, with the specific suggestion that ET datasets
could be combined on a per-biome scale, if some datasets are known to perform better or
worse under specific conditions (Miralles et al., 2016). However, this approach has the
disadvantage of obscuring the individual problems with each dataset (Miralles et al., 2016). It
may also perturb the larger-scale spatial patterns of ET. Additionally, knowledge of the
individual product errors must be well known so that uncertainty propagation and weighting is
possible. Given that the validity of the assumptions behind triple collocation are not fully
known, any such effort would benefit from additional corroboration of the estimated
uncertainties.

***4.2. Do spatial patterns in SFE performance match our expectation?***
We find that the performance of SFE is not more sensitive to biogeographical gradients than
that of other datasets, suggesting that the simplicity of SFE does not exacerbate performance
issues for specific climate, vegetation, or topographical environments. This is particularly
surprising given the previously hypothesized limitation of SFE in coastal regions, where
atmospheric conditions strongly depend on the influence of the ocean as well as on recent land
fluxes (McColl and Rigden, 2020). However, the SFE method has not previously been applied
within 250 km of the coast, let alone had its errors characterized in these regions. Therefore,
the actual performance of SFE in coastal regions has previously remained unknown.
While our statistical analysis (Figure 7, Figure 8) shows the expected increase in SFE $\sigma_\epsilon$
and reduction in $R_T$ near the coast, particularly within the first four pixels (~20 km), this
behavior is also true for ERA5-Land, which has even more severe performance decreases near
the coast than SFE. This is despite the improved simulation of land surface temperature and
surface energy fluxes in ERA5-Land compared ERA5 for coastal regions, which has been mainly
attributed to ERA5-Land's finer spatial resolution (Martens et al., 2020; Muñoz-Sabater et al.,
2021). However, ERA5-Land performance is not uniformly degraded for all coastal areas (Figure
3). Instead, coastal areas in the North show higher $\sigma_\epsilon$ and $R_T$ compared to coastal areas in the
Southwest and Southeast. This might suggest that the statistically lower performance of ERA5-



Land with coastal proximity in general is due to cross correlation with other climatic factors.
Despite the decreased performance of SFE and ERA5-Land near the coast, however, the
absolute magnitude of $\sigma_\epsilon$ and $R_T$ for both datasets is still comparable to those of the other
datasets throughout the range of coastal proximities, particularly for $\sigma_\epsilon$. Therefore, coastal
proximity may not necessarily limit the usefulness of SFE near coasts. Future SFE
implementation and evaluation studies should further investigate these limitations and not
exclude areas within 250 km of the coast *a priori.*

SFE has the highest $\sigma_\epsilon$ at low elevations, as does GLEAM and FluxCom. Spatially,
however, topographical gradients (such as around the Rocky Mountains) are not apparent on
maps of $\sigma_\epsilon$ for any of the datasets (Figure 3), although several smaller mountain ranges (e.g. the
Sierra Nevada in California and the upper Appalachian Mountains) do show lower performance
for the $R_T$ of SFE and FluxCom. This lack of coherence between the elevation trends and spatial
patterns could indicate cross correlation between elevation and other factors impacting
performance, which require further investigation.

The most obvious spatial trend in dataset performance is the gradient of performance
between the Eastern and Western US. Contrary to expectation, SFE and FluxCom have lower $\sigma_\epsilon$
in the Western US than in the East. One possible explanation for our results is that ET amounts
are lower in the West, where vegetation cover is in general lower and aridity higher, such that
the overall magnitudes of $\sigma_\epsilon$ are also lower. This would also explain the lack of systematic
difference in FluxCom and SFE $R_T$ in the East vs the West. Another explanation might be that
SFE and FluxCom both have the highest performance (for both low $\sigma_\epsilon$ and high $R_T$) in shrublands
and grassland land cover types, both of which are often found in the Western US (Dewitz,
2024). This finding is in contrast to Zhu et al. (2024), who found that daily and monthly SFE had
the lowest correlation and highest root mean squared error at the eight towers in shrublands,
relative to towers in other land covers.

### 4.3. The benefits and limitations of triple collocation

Triple collocation makes several assumptions, including that the random errors between the
datasets are independent, that the random errors are stationary across time, and that the
random errors can be described linearly. The assumptions of triple collocation are also implicitly
made by more standard validation analyses such as comparison via RMSE (Gruber et al., 2016).
However, these assumptions are expected to be violated to some degree, regardless of how
carefully comparison datasets are chosen. One reason for this is that most ET models contain at
least some overlapping input data, for example the commonly used MODIS reflectance
products for vegetation, such as leaf area index, are used as inputs to FLUXCOM, ERA5-Land,
and GLEAM (ECMWF, 2018; Jung et al., 2019; Miralles et al., 2025). Any overlap in model input



data reduces the likelihood that the resulting ET estimates will have independent errors. Triple
collocation may also fail or wrongly estimate dataset errors if random error magnitudes vary in
time or are not well described linearly. Therefore, it is not uncommon for triple collocation
studies to have invalid pixel results (e.g. He et al., 2023). Some triple collocation studies also
choose to pre-filter pixels to ensure high correlation coefficient between the raw datasets
(Gruber et al., 2016; McColl et al., 2014), which also leads to pixels where triple collocation
results are missing.
One way to increase the confidence in an application of triple collocation is to repeat
the analysis for multiple triplets, as performed here. Violations in the triple collocation
assumptions would lead to differences in the estimated error statistic for a given dataset
depending on which datasets are used for comparison (He et al., 2023; McColl et al., 2014). We
found that invalid triple collocation results were more prevalent when FluxCom and ERA5-Land
were compared within the same triplet, regardless of the third dataset. This suggests that the
assumption of independent errors may be worse between these two datasets, despite their
seemingly larger input difference than GLEAM and FluxCom, for example, which both
incorporate machine learning. Nevertheless, the overall high agreement between different
triple collocation estimates for the other triplets - and the lack of coherent spatial pattern in
error variability across triplets (Figure 5) - strongly increases our confidence that our overall
error estimates are robust.
One limitation of triple collocation is that it cannot provide information about
multiplicative dataset biases ($\beta_i$ ) beyond estimating relative biases with reference to one
member of each triplet which is assumed to have no bias (Gruber et al., 2016; McColl et al.,
2014). However, previous work suggests that SFE may have issues with bias particularly along
aridity gradients. For example, Chen et al. (2021) and Zhu et al. (2024) both found that SFE ET
had higher bias in arid conditions and tended to underestimate ET in wet conditions. This same
pattern was also observed for comparisons of in situ SFE to eddy covariance data (McColl and
Rigden, 2020; Thakur et al., 2025). While we do not consider bias because triple collocation only
allows for its calculation relative to a comparison dataset, we do see that SFE $\sigma_\epsilon$ is highest at the
driest and wettest pixels compared to pixels with intermediate mean annual precipitation. SFE
$R_T$, on the other hand, shows only a weak but slightly decreasing relationship with increasing
mean annual precipitation. Additional investigation into this is necessary. However the problem
of ET overestimation in arid conditions - when surface evaporation is high in general - is not
unique to SFE (McColl and Rigden, 2020; Miralles et al., 2016; Salvucci and Gentine, 2013).
Despite the assumptions and limitations of triple collocation, the method's ability to quantify
error statistics relative to true ET without needing an error-free dataset of ET remains a
substantial and unique benefit.



**5. Conclusions**

SFE allows for observational, data-driven estimates of ET with no tunable parameters or land surface information required. In leveraging land-atmosphere coupling, SFE estimates ET from atmospheric conditions alone, and therefore provides an opportunity to test hypotheses about vegetation response to environmental drivers without assuming that response *a priori* in the creation of the ET estimate itself. The lack of parameterization for SFE eases issues of circularity constraining research into essential outstanding challenges in ecohydrology, such as the response of ET to drought (Zhao et al., 2022) and the inference of subsurface water storage from changes in vegetation behavior (Dralle et al., 2020; Feldman et al., 2023; Stocker et al., 2023). Based on triple collocation - and despite its simplicity - SFE exhibits comparable performance to the more complicated ET estimates from GLEAM, FluxCom, and ERA5-Land.

**6. Code availability**

Code is available on GitHub at https://github.com/erica-mccormick/surface-flux-equilibrium.

**7. Data availability**

All of the data used to estimate SFE ET as well as the comparison ET datasets are publicly available online. Daily 4 km estimates of SFE ET across CONUS calculated here from 1979 to 2024 will be made available at Zenodo upon acceptance of the manuscript.

**8. Acknowledgments**

ELM, LES, and AGK were supported by NSF DEB 1942133. AGK was also supported by the Alfred P. Sloan Foundation and by the Gordon and Betty Moore Foundation under grant 11974. ELM was supported by the Stanford University Diversifying Academia Recruiting Excellence Doctoral Fellowship and by the NSF GRFP. KAM acknowledges funding from NSF grant AGS-2129576, an NSF CAREER award (AGS-2441565), and a Sloan Research Fellowship (FG-2023-19963).

**9. Competing interests**

The authors declare that they have no conflict of interest.

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

A High-Resolution Global Hydrography Map Based on Latest Topography Dataset, Water
Resour. Res., 55, 5053–5073, https://doi.org/10.1029/2019WR024873, 2019.



Yang, Y., Roderick, M. L., Guo, H., Miralles, D. G., Zhang, L., Fatichi, S., Luo, X., Zhang, Y.,
McVicar, T. R., Tu, Z., Keenan, T. F., Fisher, J. B., Gan, R., Zhang, X., Piao, S., Zhang, B., and
Yang, D.: Evapotranspiration on a greening Earth, Nat. Rev. Earth Environ., 4, 626–641,
https://doi.org/10.1038/s43017-023-00464-3, 2023.

Yilmaz, M. T. and Crow, W. T.: Evaluation of Assumptions in Soil Moisture Triple Collocation
Analysis, J. Hydrometeorol., 15, 1293–1302, https://doi.org/10.1175/JHM-D-13-0158.1,
2014.

Yin, X., Jiang, B., Liang, S., Li, S., Zhao, X., Wang, Q., Xu, J., Han, J., Liang, H., Zhang, X., Liu, Q.,
Yao, Y., Jia, K., and Xie, X.: Significant discrepancies of land surface daily net radiation
among ten remotely sensed and reanalysis products, Int. J. Digit. Earth, 16, 3725–3752,
https://doi.org/10.1080/17538947.2023.2253211, 2023.

Zhao, M., A, G., Liu, Y., and Konings, A. G.: Evapotranspiration frequently increases during
droughts, Nat. Clim. Change, 12, 1024–1030, https://doi.org/10.1038/s41558-022-01505-3,
2022.

Zhu, W., Yu, X., Wei, J., and Lv, A.: Surface flux equilibrium estimates of evaporative fraction
and evapotranspiration at global scale: Accuracy evaluation and performance comparison,
Agric. Water Manag., 291, 108609, https://doi.org/10.1016/j.agwat.2023.108609, 2024.