# Peer review of "Triple collocation validates CONUS-wide evapotranspiration inferred from atmospheric conditions"

_EGUsphere, 2025_

## Author Comment (AC1)

**Response to Alexander Gruber**

*Reviewer: This manuscript shows a rigorous analysis of the uncertainties in evapotranspiration estimates from "classical" and an alternative method using triple collocation analysis. It is very well written, clear, sound, relevant, and fits well into the scope of HESS. I recommend this manuscript to be published after minor revisions, which mainly concern clarifications of the methodology and justifications for certain assumptions.*

**Response:** We are glad the reviewer believes our study is rigorous and relevant and we appreciate the suggestions.

*My two main concerns are:*

*Reviewer: Sec. 2.2: How justified is the linear error model for ET? Given the non-linear nature of Eq. 1, I am a bit worried that it might not be. Then again, I don't know much about error structures in ET data, so this is more of a personal gut feeling. I can imagine other people having similar concerns though, so perhaps you could add some words on that, or a reference to previous work that had looked into that?*

**Response:** We agree that a linear error model may not fully capture the error structure of the ET datasets. However, a linear error model has been successfully used to evaluate ET using triple collocation in other regions (Kahn et al., 2018, He et al., 2023). We will add the following sentence to section 2.2 (line 151) to acknowledge this:

> *"While a linear error model likely does not fully capture the error structure of the actual ET dataset errors, it has been successfully used to evaluate ET datasets using triple collocation in other regions (Kahn et al., 2018, He et al., 2023)."*

One step we take toward mitigating the impact of non-linear error structures is to perform triple collocation on climatological anomalies from all datasets, rather than on the raw values. This has been shown to improve triple collocation error estimates for ET in general (He et al., 2023), but it also serves to eliminate low frequency (e.g. seasonal) ET signals which are expected to have different non-linearity than the high frequency signals which are isolated by the anomaly (Su et al., 2014, Miralles et al., 2010). We will change the following text in section 2.3 (lines 192-194, additions in **bold**) to highlight this for readers:

> *"We also remove the seasonal cycle from each dataset by subtracting the 30-day rolling average from each day (Chen et al., 2018; Draper et al., 2013; Miralles et al., 2010). This ensures that differences in the seasonality and timing of ET do not impact the triple collocation analysis*

*and has been shown to improve error estimation with triple collocation for ET datasets specifically (He et al., 2023). **Performing triple collocation on the anomaly should also reduce violation of the assumption that the ET error structure is linear. This is because the low-frequency (e.g. seasonal) ET signals which are removed are expected to have a different non-linearity than the high-frequency signals isolated by the anomaly (Miralles et al., 2010, Su et al., 2014)."***

Finally, as we note in the manuscript, our replication of the triple collocation results using different combinations of datasets supports our use of the triple collocation assumptions (including the assumption of a linear error model). We do find combinations of datasets that produce non-valid results for certain pixels, which we agree are likely due to violations of the triple collocation assumptions. However, the overall similarity in our main findings irrespective of dataset choice increases confidence that triple collocation with a linear error model continues to be useful for evaluating ET datasets.

References added to manuscript:

Su, C. H., Ryu, D., Crow, W. T., & Western, A. W. (2014). Beyond triple collocation: Applications to soil moisture monitoring. Journal of Geophysical Research: Atmospheres, 119(11), 6419-6439.

**Reviewer:** Discussion: Your discussion revolves around the different patterns you see in \sigma_eps and R_T. If I understand your your methodology correctly, you compare *unscaled* \sigma_eps estimates (Eq. 6). How meaningful is such a comparison? In Fig 2. you show clearly that the different data sets have a different mean and variability, thus we do expect variations in the \beta terms. I'm not an ET guy, but I assume that most data set applications would try to get rid of any systematic error and therefore scale the random errors accordingly. So I would argue that it only makes sense to compare scaled random error variances, i.e., \sigma_eps, that relate to the same signal variability. After all, it is the signal-to-noise ratio that determines how well the data set information can be separated from underlying noise, and this is directly reflected (in a normalized way) by R_T.

**Response:** We agree with the reviewer that if ET data applications first rescale any large-scale dataset they use, it would make sense to also do this in our application of triple collocation. However, this is rarely the case in studies using large-scale applications of ET to compare to models (e.g. Moshir Panahi et al., 2021, Xu et al., 2024), nor is it the case in studies that compare ET datasets to other remote sensing or in situ datasets of other variables (e.g. Qui et al., 2020, Zhang et al., 2023). Furthermore, even when rescaling occurs, it may not always be done simply using a

linear rescaling analogous with that in the re-scaled version of the triple collocation equations. We therefore believe it is clearest for the reader - and has the widest possible applicability - to not rescale the datasets prior to implementing triple collocation.

References:

Moshir Panahi, D., Sadeghi Tabas, S., Kalantari, Z., Ferreira, C. S. S., & Zahabiyoun, B. (2021). Spatio-temporal assessment of global gridded evapotranspiration datasets across Iran. *Remote Sensing*, 13(9), 1816.

Qiu, J., Crow, W. T., Dong, J., & Nearing, G. S. (2020). Model representation of the coupling between evapotranspiration and soil water content at different depths. *Hydrology and Earth System Sciences*, 24(2), 581-594.

Xu, C., Wang, W., Hu, Y., & Liu, Y. (2024). Evaluation of ERA5, ERA5-Land, GLDAS-2.1, and GLEAM potential evapotranspiration data over mainland China. *Journal of Hydrology: Regional Studies*, 51, 101651.

Zhang, W., Koch, J., Wei, F., Zeng, Z., Fang, Z., & Fensholt, R. (2023). Soil moisture and atmospheric aridity impact spatio-temporal changes in evapotranspiration at a global scale. *Journal of Geophysical Research: Atmospheres*, 128(8), e2022JD038046.

*Other comments:*

*__Reviewer:__ It is stated repeatedly that one advantage of the SFE method is that it doesn't make assumptions about root-zone soil moisture or vegetation status. I understand that the SFE method doesn't require one to do that directly, but aren't such assumptions necessary for the computation of air temperature and humidity that are used as input for the SFE method?*

**Response:** For general SFE applications, these assumptions may not be strictly necessary if the input humidity and temperature data comes from a source that does not have a land surface modelling component, such as from in situ stations or from station data that are interpolated using empirical/statistical methods rather than physical modelling-based methods. Thus, our statement is correct in general.

However, we agree that for the specific application here, the gridMET dataset we use for humidity and air temperature does incorporate modelling output from the North American Land Data Assimilation System (NLDAS), although it is bias corrected with in

situ station data (from PRISM). Assumptions about root-zone soil moisture or vegetation status in NLDAS could still impact the temperature and humidity estimates, however.

We will add the following to Section 2.1 (Line 141) to better explain the gridMET dataset (changes in **bold)**:

> *"We choose gridMET **because it downscales output from the North American Land Data Assimilation System (NLDAS) with PRISM. This incorporation of statistically interpolated station data at a fine resolution helps gridMET achieve a high correlation with in situ stations, particularly for the variable of temperature, while maintaining** a relatively fine spatial resolution of 4 km at a daily timescale across CONUS **(Abatzoglou, 2013)**. Net radiation (Rn) allows conversion from the Bowen ratio to ET (Eq 2). We use Rn from ERA5-Land (Muñoz-Sabater et al., 2021) because of its high agreement with in situ measurements across CONUS (Yin et al., 2023). **However, we note that error in these input datasets will propagate to error in the resulting ET estimates."***

We will also add the following text in **bold** to acknowledge the impact of dataset choice on the SFE estimates (Line 452, section 4.1):

> *"...SFE contains no a priori assumptions about the effect of water stress on ET, **aside from any impact of these assumptions embedded in the interpolated temperature or humidity data used as an input to SFE calculation (such as for the gridMET data used here)."***

**Reviewer:** *The term "error" and variations thereof are used a bit loosely. There is currently a push to harmonize the terminology concerning "errors" across communities; I recommend having a look at Merchent et al. (2017) and consider adopting their proposed terminology (in particular the usage of "error" vs. "uncertainty").*

**Response:** Thank you for this recommendation. We will eliminate our vague use of the term 'uncertainty' in the manuscript by revising Line 475 (changes in **bold**):

> *"While not often practical for large-scale use, He et al., (2023) used triple collocation to estimate an 'optimal' ET product over China by weighting each dataset by **the performance of the triple collocation results in order to minimize $\sigma_\varepsilon$."***

We will also replace our use of the word 'error' with 'uncertainty' in Line 64:

*"For example, SFE ET has been found to be within the range of in situ measurement **uncertainty** at a selection of inland eddy covariance towers, an upper limit on the performance of any ET estimate (McColl and Rigden, 2020)."*

*Reviewer:* *Sec. 2.3 I'm missing an explanation what you do with the redundant TCA estimates from the different triplets.... In the supplement, you show the results of the individual triplets, which is fine, but in the main text, it is not clear what you show... I assume it is the average of the estimates from all triplets? Did you average for both \sigma_eps and R_t? If so, it is generally advised NOT to average correlation coefficients, but this advise comes from averaging Pearson correlations; I'm not sure if that holds here too. One could actually throw in all the four data sets in a least-squares estimator to get adjusted estimates for the signal and error variance (see Gruber et al., 2016), and then derive the R_t estimates from these, which may be a bit more robust but I'm only speculating here. Anyway, I think it would be good to at least elaborate what you did/show.*

**Response:** We show the mean (and, in Figure 5, the standard deviation) of the results across the triplets. This is stated in the text as well as in the figure caption. However, we will edit the colorbar label for Figure 3 by adding the phrase "mean across triplets" to make this more clear.

[Figure]

*Figure 3. The standard deviation of the random error, σε (left) and correlation coefficient to the truth, RT (right) for each dataset averaged across all triplet combinations. Increasingly light colors are better performance. White pixels have no valid data for any triplet.*

*Reviewer: Sec. 3.2: Section titles for 3.1. and 3.3 state what its shown whereas the section title for 3.2. is a spelled out conclusion. In the discussion, titles change again to questions. I suggest just choosing one title naming style and stay consistent.*

**Response:**
We will change heading 3.2, 4.1, and 4.3 as written for formatting consistency within each main section of the manuscript - with a focus on the topic rather than the one conclusion (to allow for multiple conclusions per section). Changes are in **bold:**

3.1. Surface flux equilibrium ET across CONUS from 1979 to 202**5**
3.2. **Comparing the** standard deviation of the random error and correlation coefficient **for each dataset**
3.3. Performance across biogeographical factors
4.1. Which ET estimate is most accurate**?**
4.3. **What are** the benefits and limitations of triple collocation**?**

*Reviewer: L81-82: "This suggests that..." The use of the embedded relative clause with a dangling preposition felt a bit awkward to read, I suggest rephrasing this sentence.*

**Response:** We will rephrase as follows: *"This suggests that SFE is robust to choices of input data, at least at the scale of eddy covariance towers."*

*Reviewer: L104: As far as I know, triple collocation is similar, but not the same as the "three-cornered hat" approach (see e.g., Sjoberg et al., 2021). I recommend to just remove this parenthetical clause.*

**Response:** Thank you for pointing this out. We will remove the clause.

*Reviewer: L139: C_p here is upper case but in Eq. 1 it's lower case. Also, perhaps change all equations symbols in text to equation mode (italic) to be consistent with the Equations?*

**Response:** We will ensure that the variable case and italics are consistent between the equations and the in-text symbols, including consistent use of lower case for $c_p$.

*Reviewer: L145: Why 10%? Can you justify that number, and might it be useful to mention the implications of this assumption?*

**Response:** The ground heat flux can vary from around 10% of $R_n$ to as much as 50% of $R_n$, depending on ground cover (Clothier et al., 1986, Santanello and Friedl, 2003). Previous SFE implementations have either neglected the ground heat flux entirely (e.g.

Chen et al., 2021) or have calculated SFE with in situ data from FluxNet where estimates of the soil heat flux are available (e.g. Zhu et al., 2024).

We have performed a sensitivity analysis of SFE for values of G of 0%, 10%, 15%, and 20%. We find that - while the magnitude of daily ET is by definition impacted by the choice of G - the results from triple collocation (i.e. the error statistics of SFE) change very little. We will add the following two figures to the SI (and re-order the remaining SI figures) to show the change in mean annual ET across CONUS and the change in the mean $\sigma_\varepsilon$ and $R_T$ across triplets for various choices of G.

By definition, increasing the assumed percentage of net radiation that is partitioned to the ground heat flux reduces the magnitude of SFE ET (new Figure S1), which also reduces estimates of $\sigma_\varepsilon$. However, the choice of G has little impact on $R_T$ (new Figure S7).

These two figures will be referenced in section 3.1 and 3.2 with the following text:

> (Line 273) "*This spatial pattern exists regardless of the choice of parameter for the ground heat flux (G), although the magnitude of mean annual ET is altered (Figure S1).*"

> (Line 373) "*The triple collocation results are also relatively insensitive to the choice of the ground heat flux (G) parameter used in the calculation of SFE, although increases in G necessarily reduce ET estimates, and therefore also reduce $\sigma_\varepsilon$ (Figure S7). To the extent that uncertainty in G causes errors in the SFE ET estimate, it will also cause errors in estimates from other ET products, which must make similar assumptions or approximations for G.*"

[Figure]

*Figure S1. The difference in mean annual SFE ET from 1979 to 2025 for different values of the ground heat flux (G).*

[Figure]

*Figure S7. The change in the standard deviation of the random error ($\sigma_\varepsilon$, left column) and the correlation coefficient ($R_T$, right column) averaged across all possible triplets for SFE calculated with different values of the ground heat flux (G), expressed as a percentage of total net radiation. Grey indicates no change.*

References:

Clothier, B. E., Clawson, K. L., Pinter Jr, P. J., Moran, M. S., Reginato, R. J., & Jackson, R. D. (1986). Estimation of soil heat flux from net radiation during the growth of alfalfa. *Agricultural and forest meteorology*, 37(4), 319-329.

Purdy, A. J., Fisher, J. B., Goulden, M. L., & Famiglietti, J. S. (2016). Ground heat flux: An analytical review of 6 models evaluated at 88 sites and globally. *Journal of Geophysical Research: Biogeosciences*, 121(12), 3045-3059.

*Reviewer:* *L161: I find the explanation "By treating the product of \sigma_T as a single unknown variable ..." a bit misleading. It is not the fact that they are treated as a single variable which lets you solve for the error variance, its the fact that the betas for two data sets cancel out in the covariance ratios, which then lets you get rid of the sigma_T term by subtracting the resulting estimate from the variance of the data set.*

**Response:** We will reword this explanation as follows:

> "The $\beta_i$ and $\sigma_T$ terms cancel out for the ratio of each dataset covariance
> pair, resulting in six equations and six unknowns."

*Reviewer:* *L199: "increasing the robustness of TC assumptions" sounds a bit odd. I guess you mean that convergence of error estimates increases our confidence that the assumptions are valid?*

**Response:** We will revise the sentence as follows, adding the text in **bold:**

> "Convergence of the error estimates regardless of the triplet chosen
> increases **our confidence in the** robustness of the triple collocation
> assumptions and **therefore** in our calculated values (Draper et al., 2013;
> 200 He et al., 2023)."

*Reviewer:* *L278--: You compare the ET estimates qualitatively and mention some numbers in the text, but I think it could be useful to also show a summary table with all the relevant metrics (e.g., correlations and biases between all data set combinations).*

**Response:** We will add the following two tables to the SI showing the correlation coefficients between each dataset pair (Table S1) and showing the mean annual ET and bias (relative to SFE) for each dataset (Table S2).

*Table S1. Correlation coefficients between each ET dataset.*

|  | GLEAM | FluxCom | ERA5-Land |
|---|---|---|---|
| **SFE** | 0.55 | 0.41 | 0.43 |
| **GLEAM** | - | 0.51 | 0.20 |
| **FluxCom** | - | - | 0.71 |

*Table S2. Mean annual ET across CONUS (excluding large water bodies) for each dataset from 1979 through 2016 and mean bias relative to SFE.*

|  | Mean annual ET (mm/year) | Bias relative to SFE |
|---|---|---|
| **SFE** | 538 | 1.0 |
| **GLEAM** | 552 | 1.03 |
| **FluxCom** | 609 | 1.13 |
| **ERA5-Land** | 645 | 1.20 |

*Reviewer:* L421: The acronym MAP hasn't been introduced.

**Response:** We will replace the acronym MAP with "mean annual precipitation" in the four places where it is used.

*Reviewer:* L472-482: Doing a weighted averaging comes from least squares theory and serves the purpose of reducing random errors only. I guess what is meant with "this approach has the disadvantage of obscuring the individual problems" is that if data sets have different systematic errors, especially if they are non-stationary, then you create some uncontrolled blend of biased estimates, and any improvement is only a matter of luck because weights derived from random error variances do not account for these biases that are instead assumed to be zero.

**Response:** Agreed. We will add the following (changes in bold) to further explain this point:

> "However, this approach has the disadvantage of obscuring the individual problems with each dataset, **especially if the datasets have different systematic errors or biases which are not accounted for by the random error variance and correlation coefficient metrics available through triple collocation analysis.**"

*Reviewer:* L482-484: Isn't this statement trivial and already implied by the paragraph's introductory statement: "It is possible to average ET estimates weighted by each dataset's performance"?

**Response:** We will remove the following sentence which we agree is redundant:

> "Additionally, knowledge of the individual product errors must be well known so that uncertainty propagation and weighting is possible."

*Reviewer:* *L521: Why is this contrary to expectation? You do state that this might have to do with the lower ET amounts in these regions, so considering my argument in the beginning concerning scaling in TCA, I would argue that this is simply a result of showing unscaled \sigma_eps estimates. When looking at signal-to-noise ratios instead, this gradient vanishes, right?*

**Response:** We agree this is a good explanation of this pattern and make a similar case in the sentence following the one referenced in this comment. However, to make this more clear, we will revise as follows, adding the text in bold:

> *"SFE and FluxCom have lower $\sigma_\varepsilon$ in the Western US than in the East,* ***despite the Western US being well-known as a region where ET estimation is difficult.*** *One possible explanation for these results is that ET amounts are lower in the West, where vegetation cover is in general lower and aridity higher, such that the overall magnitudes of $\sigma_\varepsilon$ are also lower."*

*Reviewer:* *L589: "complex" instead of "complicated"?*

**Response:** Agreed.

*Reviewer:* *Eq. (4)-(7): The introduction of Q_ii seems a bit unnecessary to me. Since you define Q_ii just as equivalent to \sigma^2_ii, you could use the latter instead of Q directly in Eqs. 6 and 7, which I don't think would make it any more difficult to read. This might be just a personal preference though.*

**Response:** We chose to follow the identical notation and formulation of the equations in McColl et al., 2014 for consistency, although we agree the Q notation is not strictly necessary here.

*Reviewer:* *Figure 1: The x-axis date labelling confused me when I first looked at it. The figure caption only states "Mean annual SFE from 1979 to 2024"... Perhaps also spell out the date range shown in the example time series: "Points show time series for [...] from Dec. 2000 to Dec. 2002"?*

**Response:** Thank you. We have relabelled the x-axis, added a title ("example pixels") above the subplots, and edited the caption, adding the bolded text:

[Figure]

*Figure 1. Mean annual SFE ET across CONUS from 1979 to 2024. Points show timeseries for example pixels **from Dec. 2000 to Dec. 2002** for SFE (green), ERA5-Land (blue), GLEAM (purple), and FluxCom (pink).*

**Reviewer:** *Figure 7/8: The order of the Figure panels is inconsistent.*

**Response:** Thank you for pointing this out. The subplots will be re-arranged so that the order matches between figures and with the order of discussion in the text:

[Figure]

*Figure 7: The standard deviation of the random error, σε, for each ET dataset across the distance to large water bodies, elevation, mean annual precipitation, and land cover. The number of pixels in each category per ET dataset is shown below boxes.*

[Figure]

*Figure 8: The correlation coefficient, $R_T$, for each ET dataset across the distance to large water bodies, elevation, mean annual precipitation, and land cover. The number of pixels in each category per ET dataset is shown below boxes.*

**Reviewer:** *Supplement: I always find it hard to visually compare patterns like these. You draw the conclusion that differences are small when using different triplets, therefore assumptions can be considered to be valid. But when exactly are differences "small enough" to draw this conclusion? There isn't an awful lot of contrast in the figures, and there indeed seem to be regions with some greater differences. Perhaps it might be worth plotting the actual \*differences\* between the TCA results for triplet combinations, or maybe complement the maps you show with boxplots of the differences?*

**Response:** We will add the following two figures to the SI to show the difference between each possible triplet pair for each dataset:

[Figure]

*Figure S4. The difference in the random error standard deviation ($\sigma_\varepsilon$) between pairs of triplets (columns) for each dataset (rows). White areas are those where the random error standard deviation can not be evaluated because of negative standard deviation, indicating that the assumptions of triple collocation are not met.*

[Figure]

*Figure S6. The difference in the correlation coefficient with the truth ($R_T$) between pairs of triplets (columns) for each dataset (rows). White areas are those where the correlation coefficient is greater than one, indicating that the assumptions of triple collocation are not met.*

References:

https://doi.org/10.5194/essd-9-511-2017

https://doi.org/10.1175/JTECH-D-19-0217.1

https://doi.org/10.1002/2015JD024027

---

## Author Comment (AC2)

**Response to Reviewer #2**

**Summary and significance**

*Reviewer: This manuscript fits well within the scope of Hydrology and Earth System Sciences. The authors introduce a new daily, 4km evapotranspiration (ET) dataset over continental US (CONUS) using the surface flux equilibrium (SFE) approach and compares it against other ET products (GLEAM, FluxCom, ERA5-Land). The authors present a careful statistical evaluation via triple collocation, giving random error and correlation to truth metrics. This manuscript is well written and conceptually clear. I particularly appreciate how the authors have put great effort and care in explaining how SFE compares to other ET estimation approaches and explains assumptions, strengths and limitations.*

*The demonstration that SFE has comparable performance to more complex approaches in many regions, particularly in the western US, is useful as it adds confidence in SFE as a practical alternative to estimate ET.*

*I recommend publication with minor revisions for clarity.*

**Response:** Thank you!

*Suggestions*

*Reviewer: Figure 1: Clarify what panels b-g are showing by giving them a title and explicitly labeling the x-axis. The current x-axis was confusing, and I suggest writing out the month and year (e.g., Dec 2000).*

**Response:** We will relabel the x-axis, add a title ("example pixels") above the subplots, and edit the caption as follows, adding the bolded text:

[Figure]

*Figure 1. Mean annual SFE ET across CONUS from 1979 to 2024. Points show timeseries for example pixels **from Dec. 2000 to Dec. 2002** for SFE (green), ERA5-Land (blue), GLEAM (purple), and FluxCom (pink).*

*Methods*:

*Reviewer: L141-144: I understand that the input data for SFE has been proven to be robust at the eddy covariance tower level (addressed in the introduction, Thakur et al., 2025). This may be extended when using gridMET and ERA5-Land data for this analysis, but can the authors directly tie that logic in Section 2.1? Can the authors address the biases of gridMET and ERA5-Land and how that may affect SFE ET?*

**Response:** We will add the following (changes in **bold**) to section 2.1:

> *"We choose gridMET **because it downscales output from the North American Land Data Assimilation System (NLDAS) with PRISM. This incorporation of statistically interpolated station data at a fine resolution helps gridMET achieve a high correlation with in situ stations, particularly for the variable of temperature, while maintaining** a relatively fine spatial resolution of 4 km at a daily timescale across CONUS **(Abatzoglou, 2013)**. Net radiation (Rn) allows conversion from the Bowen ratio to ET (Eq 2). We use Rn from ERA5-Land (Muñoz-Sabater et al., 2021) because of its high agreement with in situ measurements across CONUS (Yin et al., 2023). **However, we note that error in these input datasets will propagate to error in the resulting ET estimates."***

*Reviewer: L145: Can the authors justify the 10% ground heat flux (G) assumption with a citation or provide a sensitivity analysis showing how varying G can affect $\sigma\varepsilon$ and RT? The former is more reasonable to accomplish, but I would want to know the authors expect $\sigma\varepsilon$ and RT to change if G is varied (e.g., 5-20%)*

**Response:** The ground heat flux can vary from around 10% of Rn to as much as 50% of Rn, depending on ground cover (Clothier et al., 1986, Santanello and Friedl, 2003). Previous SFE implementations have either neglected the ground heat flux entirely (e.g. Chen et al., 2021) or have calculated SFE with in situ data from FluxNet where estimates of the soil heat flux are available (e.g. Zhu et al., 2024).

We have performed a sensitivity analysis of SFE for values of G of 0%, 10%, 15%, and 20%. We find that, while the magnitude of daily ET is by definition impacted by the choice of G, the results from triple collocation (i.e. the error statistics of SFE) change very little. We will add the following two figures to the SI (and re-order the remaining SI figures) to show the change in mean annual ET across CONUS and the change in the mean $\sigma_\varepsilon$ and $R_T$ across triplets for various choices of G.

By definition, increasing the assumed percentage of net radiation that is partitioned to the ground heat flux reduces the magnitude of SFE ET (new Figure S1), which also reduces estimates of $\sigma_\varepsilon$. However, the choice of G has little impact on $R_T$ (new Figure S7).

These two figures will be referenced in section 3.1 and 3.2 with the following text:

> (Line 273) "*This spatial pattern exists regardless of the choice of parameter for the ground heat flux (G), although the magnitude of mean annual ET is altered (Figure S1).*"

> (Line 373) "*The triple collocation results are also relatively insensitive to the choice of the ground heat flux (G) parameter used in the calculation of SFE, although increases in G necessarily reduce ET estimates, and therefore also reduce $\sigma_\varepsilon$ (Figure S7). To the extent that uncertainty in G causes errors in the SFE ET estimate, it will also cause errors in estimates from other ET products, which must make similar assumptions or approximations for G.*"

[Figure]

*Figure S1. The difference in mean annual SFE ET from 1979 to 2025 for different values of the ground heat flux (G).*

[Figure]

*Figure S5. The change in the standard deviation of the random error ($\sigma_\varepsilon$, left column) and the correlation coefficient ($R_T$, right column) averaged across all possible triplets for SFE calculated with different values of the ground heat flux (G), expressed as a percentage of total net radiation. Grey indicates no change.*

Additionally, while conducting this sensitivity analysis, we realized that the figures we show in the manuscript as submitted were actually calculated using a G of 0%, not a G of 10% as stated. We will update all of the figures and numerical results in the text to show results for G of 10%. Because the change in TC results is minimal (see Figure S5 above), changing the choice of G has no impact on the main findings of the paper. The change is visually detectable in Figure 1 (see new version in response above), Figure 2, and Figure 4 as well as in the specific pixel counts listed in Table 1. However, the main messages of these figures remain the same.

[Figure]

Figure 2. Interannual variability in mean annual ET across CONUS from 1979 through the record length of each dataset.

[Figure]

*Figure 4. Summary of relative performance of all four datasets. The dataset with highest performance for the standard deviation of the random error, $\sigma_\epsilon$ (a) and the correlation coefficient with 'true' ET, $R_T$ (b) for each pixel. The worst performing datasets for $\sigma_\epsilon$ (c) and $R_T$ (d). The relative ranking of SFE for $\sigma_\epsilon$ (e) and $R_T$ (f). The total number of pixels (and relative percent of pixels) of each color are shown in Table S1. Pixels with centroids within 4 km of the border have been removed.*

Table 1. (Top) The number of pixels where each dataset has the best performance according to the standard deviation of the random error, σε, and the correlation coefficient to the truth, $R_T$. (Bottom) The number of pixels by SFE ET ranking.

| Best dataset | | | | |
|---|---|---|---|---|
| | By $\sigma_\varepsilon$ | | By $R_T$ | |
| | Pixels | Percent | Pixels | Percent |
| SFE | 164 | (5.4%) | 115 | (3.8%) |
| GLEAM | 17 | (0.6%) | 159 | (5.2%) |
| FLUXCOM | 2537 | (83.7%) | 33 | (1.1%) |
| ERA5-Land | 314 | (10.4%) | 2725 | (89.9%) |
| Ranking of SFE | | | | |
| | By $\sigma_\varepsilon$ | | By $R_T$ | |
| | Pixels | Percent | Pixels | Percent |
| 1st | 111 | (3.7%) | 156 | (5.1%) |
| 2nd | 1286 | (42.4%) | 2206 | (72.8%) |
| 3rd | 1397 | (46.1%) | 646 | (21.3%) |
| 4th | 238 | (7.8%) | 24 | (0.8%) |

References:

Clothier, B. E., Clawson, K. L., Pinter Jr, P. J., Moran, M. S., Reginato, R. J., & Jackson, R. D. (1986). Estimation of soil heat flux from net radiation during the growth of alfalfa. *Agricultural and forest meteorology*, 37(4), 319-329.

Purdy, A. J., Fisher, J. B., Goulden, M. L., & Famiglietti, J. S. (2016). Ground heat flux: An analytical review of 6 models evaluated at 88 sites and globally. *Journal of Geophysical Research: Biogeosciences*, 121(12), 3045-3059.

*Reviewer:* *L145: Can the authors explain how including days with negative net radiation (Rn) can affect daily ET estimation and triple collocation statistics and justify their exclusion?*

**Response:** As stated in line 145, we do not evaluate SFE ET on any day with negative net radiation. We will further explain this by adding the following in **bold:**

> *"We assume a ground heat flux (G)* **that is** *10% of* **$R_n$. Additionally,** *we do not evaluate SFE ET on any days with negative $R_n$* **because doing so would result in a negative ET estimate, which is not physical."**

We also thank the reviewer for helping us realize that we did not explicitly address how we deal with these no-data days in the triple collocation analysis. We will add the following text to Line 192 (section 2.3) to make it clear that we did not perform triple collocation on winter days, when net radiation is most likely to be negative:

> *"After removing the seasonal cycle, we choose only the months of March through October for the triple collocation analysis. This is because negative daily net radiation occurs for some pixels during the winter months, prohibiting the calculation of SFE. Because the number of days with negative net radiation varies for each pixel, we eliminate all winter months for all datasets to ensure a consistent number of data at each dataset and pixel."*

To remind the readers of this important detail, we will also reference it throughout the Results and Discussion sections, for example by adding the following text in **bold**:

> *Line 310 (Section 3.2): "SFE performance* **during non-winter months** *as estimated by triple collocation is comparable - and even exceeds - the performance of the comparison datasets across much of CONUS, despite its extreme simplicity, lack of tunable parameters, and relatively small number of assumptions (Figure 3)."*

> *Line 442 (Section 4.1): "While triple collocation reveals that SFE is rarely the highest performing dataset* **for the non-winter months evaluated in this study,** *it is the second-best performing dataset across much of CONUS for both $\sigma_\varepsilon$ and $R_T$ (Figure 4e,f)."*

*Discussion:*

*Reviewer:* *I suggest adding a brief discussion about expected SFE performance outside CONUS and considerations for global implementation. How do the authors expect SFE to perform in regions with weaker land-atmosphere coupling (e.g., Southeast Asia)?*

**Response:** There is no reason to believe that SFE should not perform similarly at the global scale, particularly outside of regions with substantial influence from ocean dynamics (such as island or coastal regions). However, global implementation is dependent on input data quality (with corresponding choices for spatial scale, for example), which is why we chose to focus this analysis on CONUS.

With regards to your question about land-atmosphere coupling: If by 'land-atmosphere coupling' we mean the feedback of not just the land on the atmosphere, but also the atmosphere on the land, then the strength of this coupling should have no impact on SFE performance. SFE does not actually take advantage of land-atmosphere coupling, but rather relies on the fact that fluxes on the land surface *do* impact atmospheric conditions (regardless of 'coupling strength'). In other words, it is not necessary for atmospheric conditions to impact surface fluxes in order for the SFE method to work.

To reduce confusion about this, we will remove the phrase "In leveraging land-atmosphere coupling" from Line 556 in the Conclusion. We will also edit Lines 556-560 in the Conclusion (adding the text in **bold**) to reiterate the possibility of estimating ET beyond CONUS:

> *"**That** SFE estimates ET from atmospheric conditions alone **has several advantages: It can be calculated at a variety of scales and geographic domains and** it provides an opportunity to test hypotheses about vegetation response to environmental drivers without assuming that response a priori in the creation of the ET estimate itself."*

*Reviewer:* *L572: The authors note that SFE bias in arid conditions needs further investigation. Can the authors please add specific recommendations for future investigation and/or what additional measurements may be needed (advocating for certain measurements?).*

**Response:** We will replace the sentence "*Additional investigation into this is necessary*" with "*Further in situ validation of SFE in arid ecosystems in particular would be beneficial.*" (Line 578)